# A study on the relationship between college students' physical exercise and feelings of inferiority: The mediating effect of social support

Huan Lei[1], Bo Peng[2], Minghuan Tang[3], Weisong Chen[2], Hongshen Wang  [2] *, Ting Yu[4]

**1** School of physical education, Chengdu Sport University, Chengdu, Sichuan, China, **2** School of sports training, Chengdu Sport University, Chengdu, Sichuan, China, **3** School of History and Culture, Chengdu Sport University, Chengdu, Sichuan, China, **4** Jingshan Primary School, Changshou, Chongqing, China

* hongshen1101@163.com

## Abstract

### Objective

This study explores the relationships among physical exercise, social support, and feelings of inferiority among college students, with a particular focus on the mediating role of social support. Using a sample of Chinese college students, the results demonstrate that physical exercise significantly reduces feelings of inferiority, both directly and indirectly through enhanced social support.

### Method

Pearson correlation analysis confirmed the positive correlation between physical exercise and social support, while also showing that both physical exercise and social support negatively correlate with feelings of inferiority. Structural equation modeling further supported the hypothesized relationships, revealing that the mediation pathway through social support accounted for 51.95% of the total effect. Additionally, multi-group invariance testing indicated that these relationships were consistent across genders, underscoring the universal applicability of the model.

### Results

The findings highlight the dual pathways through which physical exercise alleviates feelings of inferiority: by fostering personal competence and enhancing social resources.

### Conclusion

This study emphasizes the importance of integrating physical exercise and social support, providing evidence-based intervention recommendations for improving students' mental health. By validating the dual-pathway model, the research offers

**Data availability statement:** The original data of this study has been uploaded to the Figshare platform under the title "How psychological resilience shapes adolescents' sports participation: the mediating effect of exercise motivation" and can be accessed via DOI: https://doi.org/10.6084/m9.figshare.28588571.

**Funding:** The author(s) received no specific funding for this work.

**Competing interests:** The authors have declared that no competing interests exist.

theoretical support and practical significance for promoting students' mental health through physical exercise and social support.

---

# 1 Introduction

In Modern Higher Education, Physical exercise in modern higher education is not merely a means to enhance physical fitness but also an effective approach to cultivating psychological resilience and promoting social adaptation. With the growing emphasis on mental health in society, the psychological benefits of physical exercise have increasingly become a focus of academic research. Recent studies have revealed that the positive effects of physical exercise on mental health extend far beyond simple mood enhancement. It plays a unique role in shaping self-awareness, improving social relationships, and fostering self-esteem [1,2]. Among these issues, feelings of inferiority, as a typical negative emotional state often stemming from low self-evaluation, have a profound adverse impact on university students' learning, social interactions, and daily lives [3]. Thus, understanding how physical exercise can alleviate feelings of inferiority has become a critical topic in psychological and sports education research.

Studies have shown that physical exercise not only directly improves physical fitness but also offers participants a rich social platform [4]. In team sports such as basketball or soccer, students engage in collaboration and interaction with teammates, gradually building trust and a sense of belonging. Within such interactive environments, encouragement and support from team members help individuals gain social recognition, enhance self-assessment, and reduce negative emotions stemming from isolation [5]. The social support formed during sports activities not only fulfills the need for belonging but also plays a vital role in alleviating loneliness and boosting self-confidence [6]. Particularly in team sports, individuals are more likely to reassess their value and capabilities through positive feedback from peers, thereby mitigating excessive self-doubt [7].

Additionally, social support, as a multi-dimensional psychological resource, is indispensable in fostering individual mental health. Studies suggest that various forms of social support—such as emotional support, practical assistance, and informational support—provide comprehensive psychological backing [8]. Specifically, when facing academic pressure or life challenges, support from family, friends, and peers can significantly alleviate emotional stress and motivate individuals to confront difficulties [9]. In the context of physical exercise, this social support is embodied through guidance from coaches, encouragement from teammates, and recognition from spectators, which gradually permeates individuals' emotions and cognition. Through constant interaction, they gain self-affirmation and social acceptance [10]. This psychological encouragement and positive feedback help students find solace and recognition amidst feelings of inferiority, providing a solid foundation for their psychological development.

In summary, physical exercise serves not only as a means of physical training but also as a source of psychological nourishment and a foundation for building

support networks. Through physical exercise, university students experience positive changes in emotions, cognition, and behaviors. These changes manifest not only in the enhancement of self-worth but also in the reduction of feelings of inferiority and the steady improvement of mental health. Exploring this process offers a new perspective for psychological interventions in higher education and provides scientific evidence for designing more targeted campus sports programs and psychological support strategies in the future.

## 2 Literature review and research hypotheses

### 2.1 The impact of college students' physical exercise on feelings of inferiority

The positive impact of physical exercise on mental health has been a long-standing research focus in the field of sports psychology. A review of relevant studies both domestically and internationally reveals two primary areas of emphasis: first, exploring the relationship between physical exercise and mental health indicators, such as emotional regulation, self-concept, cognitive processes, and personality traits; and second, utilizing physical exercise as an intervention to improve psychological disorders, including but not limited to autism, anxiety disorders, depression, and schizophrenia [11–13]. Physical exercise is not only a means to promote physical health but also an important tool for improving mental well-being. Among these studies, self-concept, as a key indicator of mental health, serves as a foundational theoretical basis for understanding the psychological benefits of physical exercise. Self-concept refers to an individual's subjective evaluation of their own physical, psychological, cognitive, and emotional attributes, formed through a cumulative process of self-perception [14]. Since physical exercise is not only a form of bodily activity but also closely linked to specific social contexts, individuals can enhance their physical fitness while improving their ability to adapt to social environments through exercise. Research indicates that physical exercise enhances bodily confidence, helps individuals gain greater social recognition, and ultimately improves self-evaluation [15]. Garn A. C. and colleagues further noted that physical activities significantly enhance mental health by improving individuals' self-concepts, showcasing the multidimensional psychological benefits of physical exercise [16].

Feelings of inferiority are a negative manifestation of self-concept, defined as a negative emotional state arising from an individual's low self-evaluation [17]. When persistent, this state can lead to anxiety, social withdrawal, depression, and even severe psychological disorders [18]. The formation of feelings of inferiority is closely related to individual cognition, social comparison, and environmental support. Given that physical exercise significantly improves self-perception, enhances social support, and boosts positive emotions, it is regarded in academic circles as an effective intervention for inferiority. In recent years, researchers have conducted extensive intervention studies targeting college students and found that physical exercise positively affects feelings of inferiority on multiple levels. Studies indicate that moderate-intensity physical exercise can significantly reduce college students' feelings of inferiority while improving their physical self-esteem and psychological capital [19,20]. Through regular and moderate physical activities, college students can gradually enhance positive perceptions of their physical appearance, derive a sense of achievement from improved physical fitness, and ultimately develop a more favorable overall self-evaluation. These changes not only promote mental health but also help individuals better adapt to academic and social environments. In competitive sports contexts, college students gain confidence, pride, and a sense of achievement through competition and interaction with others [21]. These positive experiences help mitigate feelings of inferiority to some extent. In particular, participants in team sports not only find value in their roles during the activity but also strengthen their self-identity through collective support and recognition.

The form of physical exercise significantly influences its intervention effects. Studies have shown that guided physical exercise has a greater impact than autonomous exercise [22]. Guided exercise, supported by professional trainers, provides technical guidance and psychological encouragement, enabling individuals to achieve their exercise goals more effectively while feeling external attention and support, thereby significantly boosting self-confidence [23]. Additionally, team sports are

more effective in alleviating feelings of inferiority than individual sports. In team activities, individuals enhance their sense of belonging through cooperation with others and improve their self-evaluation through group recognition [24,25]. Exercise volume is significantly negatively correlated with levels of inferiority, indicating that higher exercise volume corresponds to lower levels of inferiority [19]. This finding highlights the importance of exercise duration and frequency in the intervention process. Adequate exercise not only contributes to physical health but also accumulates psychological benefits, leading to deeper emotional improvements. This provides scientific guidance for college students to design reasonable exercise plans.

Based on this understanding, we propose the following hypothesis

**Hypothesis1(H1):** Physical exercise has a direct negative effect on feelings of inferiority among college students.

## 2.2 The relationship between college students' physical exercise and social support

Physical exercise is an activity that combines collectivity and interaction, offering a unique social context that fosters the acquisition of social support. In team sports (such as basketball and soccer) or guided activities (such as fitness classes), participants often find themselves in high-frequency interactive settings where pursuing common goals strengthens emotional bonds and a sense of collaboration. These interactions not only enhance the quality of relationships but also promote a sense of team belonging [26–28]. For example, college students engaging in group sports activities often experience support and encouragement from their teammates. This support manifests not only in the shared effort on the field but also in practical help and emotional encouragement among team members [29]. Additionally, sports create unique social dynamics: competitive scenarios (e.g., matches) intensify the frequency and depth of interactions, while cooperative activities (e.g., group hiking) reinforce trust and intimacy within teams [30]. These interactions fostered by physical activities provide immediate social satisfaction while laying the groundwork for long-term interpersonal networks, offering sustainable social support for college students.

Physical exercise is not merely a physical activity but also a psychological process that enhances emotional connections. Within sports settings, participants often receive emotional support through encouragement, recognition, and companionship. Studies suggest that sports provide opportunities for emotional exchange, whether in the joy of success or the setbacks of failure. This shared resonance, from teammates, coaches, and even spectators, serves as a vital psychological driver for overcoming challenges and fostering personal growth [31]. Campus sports clubs or class activities create opportunities for shared experiences among college students. In these activities, students not only build physical fitness but also form deep friendships through emotional exchanges while confronting challenges together [32,33]. Such emotional support is crucial in alleviating loneliness and enhancing a sense of belonging, particularly in the high-pressure academic environment where sports activities offer a meaningful outlet for emotional expression and understanding [34]. The encouragement and support from team members help release tension, enhance psychological comfort, and deepen and broaden social connections. These emotional bonds extend beyond the sports setting into daily life, enabling college students to face academic and social challenges with greater ease [35].

Physical exercise also indirectly enhances social support by expanding social networks and increasing social recognition. Research indicates that participants in sports activities are more likely to interact with diverse social groups, including individuals from different ages, cultures, and backgrounds [36]. Through these interactions, participants gain access to richer resources of support, such as emotional comfort, information sharing, and practical advice for solving problems. This diversified network creates a more inclusive support system for college students, helping them establish a sense of identity and belonging in broader social contexts, thereby alleviating feelings of isolation. Additionally, participation in sports enhances an individual's social image and status. Outstanding performance in team sports or contributions to the group enable college students to earn respect from their peers for their athletic abilities or leadership skills [37,38]. This social recognition further strengthens the perception and stability of social support. For instance, exceptional performance in group sports not only brings short-term satisfaction but also elevates an individual's long-term status within the group, making it easier for them to gain support and recognition in future social interactions [39].

Based on this understanding, we propose the following hypothesis

**Hypothesis2(H2):** Physical exercise positively influences social support among college students.

## 2.3 The relationship between college students' social support and feelings of inferiority

Social support is widely recognized as a vital resource for individual mental health, playing a significant role in mitigating negative emotions [40,41]. Feelings of inferiority, as a typical negative emotional state, often stem from an individual's negative self-perception and are accompanied by psychological distress such as loneliness and helplessness. Social support, through emotional comfort and encouragement, effectively alleviates these negative emotions and injects psychological resilience into individuals [42]. Research highlights emotional support as the cornerstone of the social support system. Particularly for college students facing academic pressure or social setbacks, care from family, friends, or peers can significantly enhance psychological resilience [43]. For instance, when a student falls into self-doubt after failing an exam, encouragement from friends or understanding from family can help rebuild their confidence. Such emotional intervention not only alleviates the psychological stress caused by feelings of inferiority but also provides a source of strength to face challenges [44]. Moreover, a strong sense of social support helps break the vicious cycle of inferiority by liberating individuals from negative emotions and enabling them to view themselves with a more positive mindset [45,46].

The root of feelings of inferiority lies not only in emotional distress but also in an individual's low self-evaluation. Social support plays a critical role in reshaping self-perception through positive feedback and subtle influences. It helps individuals redefine their self-worth, gradually weakening their negative self-evaluations, thereby reducing the intensity of feelings of inferiority [47]. For example, peer support, through praise and affirmation, significantly enhances students' self-efficacy, making them more aware of their strengths [48–50]. Family support, on the other hand, provides a foundation of positive values through long-term stable relationships. Studies show that affirmation from close friends and classmates effectively prevents students from excessively magnifying their weaknesses, helping them avoid falling into cognitive traps of self-deprecation [51,52]. Additionally, social support not only fosters positive self-perception but also encourages students to actively participate in social and academic activities, further boosting their confidence. Positive feedback from participation reinforces their self-evaluation, ultimately reducing the occurrence of feelings of inferiority [53].

Beyond directly influencing emotions and cognition, social support indirectly enhances individuals' social adaptability through behavioral shaping, further reducing feelings of inferiority. Individuals with adequate social support typically have more opportunities to engage in social activities, providing them platforms to showcase themselves and accumulate successful experiences, thus creating a positive feedback loop [54–56]. For example, social support in team activities encourages students to actively engage in more social behaviors, enhancing their social adaptability through successful interactions [57,58]. Such experiences not only help overcome social anxiety but also diminish avoidance tendencies caused by feelings of inferiority. Furthermore, social support offers external resources for coping with setbacks. When students encounter failures in academics or life, practical assistance and emotional support from others significantly alleviate feelings of frustration, preventing them from falling into a vicious cycle where failure exacerbates inferiority [59,60]. Particularly in campus environments, diverse forms of social support, through varied interactive scenarios, enhance students' psychological flexibility and emotional stability, providing a solid foundation for improving their social adaptability.

Based on this understanding, we propose the following hypothesis

**Hypothesis3(H3):** Social support has a negative effect on feelings of inferiority among college students.

## 2.4 The mediating role of social support in the relationship between physical exercise and feelings of inferiority among college students

Physical exercise is not merely a means to enhance physical fitness but also a powerful catalyst for social interaction. Through participation in sports, particularly team-based activities such as basketball and soccer, college students forge meaningful interpersonal connections in contexts of collaboration and competition. These interactions extend beyond

on-field cooperation and encouragement to include the mutual support and understanding fostered during daily training sessions [61]. For instance, trust among teammates, guidance from coaches, and recognition from spectators provide participants with a sense of emotional belonging and acceptance. This sense of belonging helps reduce the feelings of inferiority caused by isolation, allowing students to discover their value within a group [27,62]. Through sports, college students gain more opportunities to express themselves and experience group support and encouragement.

Social support alleviates feelings of inferiority through emotional care and positive feedback at multiple levels. On an emotional level, friendships and care are direct manifestations of social support. When college students encounter setbacks or failures in sports, the comfort and encouragement provided by teammates offer psychological solace, enabling them to face difficulties with greater ease [43,63]. For example, a student participating in a group activity gains support and motivation from peers, which not only eases the loneliness stemming from failure but also strengthens their sense of belonging within the team. On a cognitive level, social support reshapes self-evaluation, helping individuals view their self-worth from a more positive perspective [64]. For instance, recognition from teammates, praise from coaches, and applause from spectators are forms of positive feedback that encourage individuals to reassess their abilities and value [48,49,65]. These seemingly small actions subtly transform participants' cognitive patterns, shifting their focus away from their weaknesses toward their strengths and potential. This cognitive shift significantly reduces feelings of inferiority, providing students with the psychological drive to confront life's challenges.

The influence of physical exercise extends beyond physical enhancement or short-term achievements. Its deeper significance lies in constructing a sustainable social support network for college students. Within this network, students not only receive emotional encouragement from teammates but also earn societal recognition for their contributions through their performance in sports [66,67]. For example, when students achieve success in team collaborations or excel in individual events, they often receive group admiration for their contributions. This positive experience further strengthens their social adaptability [68]. More importantly, this social network fosters healthy interpersonal relationship patterns. The positive interactions formed through sports often carry over into daily life, providing students with ongoing psychological support [69]. These relationships offer a sense of inner security and satisfaction, enabling them to gradually overcome excessive self-doubt and self-criticism. This transformation makes physical exercise an effective tool for psychological adjustment, with value extending beyond the sports field to deeply influence individual mental health and social adaptation.

### 2.5 Construction of the comprehensive theoretical hypothesis model

Based on the literature review and the related hypotheses, this study has constructed a theoretical hypothesis model, as shown in Fig 1. The model aims to integrate existing research findings, optimize the relationships among the hypotheses, and expand the theoretical framework to provide a more comprehensive understanding of the research topic.

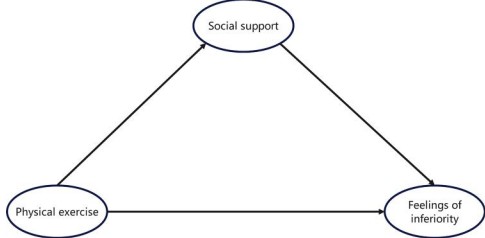

**Fig 1. Heoretical Model of the Relationship Between Physical Exercise and Feelings of Inferiority in College Students: The Mediating Role of Social Support.**

# 3 Materials and methods

## 3.1 Research participants and data

### 3.1.1 Sample size justification.
To ensure the scientific rigor and reliability of this study, we utilized both G*Power software for power analysis and classical heuristic rules in social science research to perform a detailed sample size estimation. This dual approach ensured the robustness of the sample design and the reliability of the study results.

First, using G*Power 3.1 software, a power analysis was conducted for multiple regression analysis, focusing on four independent variables: physical exercise, social support, emotional regulation ability, and feelings of inferiority. Assuming a medium effect size ($f^2 = 0.15$), a significance level of $\alpha = 0.05$, and statistical power of $1 - \beta = 0.80$, the analysis indicated a minimum sample size of 90 participants to detect significant statistical effects. This result provides a baseline for the minimum sample size required.

To further validate the adequacy of the sample size, we adopted a commonly accepted heuristic rule in social science research, which recommends that the sample size be 10–15 times the number of items in the questionnaire. This study involves three scales: the physical exercise scale (3 items), the social support scale (12 items), and the feelings of inferiority scale (36 items), for a total of 51 items. According to this rule, the minimum sample size should range between 510 and 765 participants ($51 \times 10$–$51 \times 15$), sufficient to meet the statistical requirements of the questionnaire design.

Ultimately, to ensure the broad representativeness of the study and the robustness of its conclusions, we far exceeded the minimum sample size requirements during data collection, obtaining 2,036 valid questionnaires from college students. This sample size not only significantly surpasses the G*Power power analysis minimum requirement (90 participants) but also far exceeds the upper limit of the heuristic rule (765 participants). Such a large sample size enhances the statistical power of the analysis, reduces the likelihood of sampling errors, and significantly improves the generalizability and external validity of the study findings.

### 3.1.2 Participant selection process.
To ensure representation across different regions and demographic groups, this study employed a stratified random sampling technique to minimize selection bias. The sampling framework included college students from 15 provinces and municipalities in China. The randomization process was conducted in multiple stages to achieve diversity and balance in the sample.

**Stratification by region:** First, China was divided into five geographic strata based on its socioeconomic and cultural diversity. Three provinces or cities were randomly selected from each stratum to ensure balanced regional representation (see Table 1 for details of the regions and selected provinces). Next, universities within the selected provinces or cities were randomly sampled, with the number of universities selected adjusted proportionally according to the distribution of universities in each region. This ensured that the sample accurately reflected the overall characteristics of college students in different regions. Finally, students were randomly selected from the chosen universities, ensuring that every student had an equal chance of participating in the study.

**Random selection of universities:** Within each geographic stratum, universities were selected using a random number generator. The number of universities chosen was proportional to the region's share of the national college student population, ensuring balanced regional representation. For example, if a region accounted for 20% of the total college student

**Table 1. Stratification by region and province in the sampling frame.**

| Region | Province |
|---|---|
| Eastern China | Shanghai, Jiangsu, Shandong |
| Central China | Henan, Hubei, Hunan |
| Western China | Sichuan, Guizhou, Chongqing |
| Southern China | Guangdong, Guangxi, Hainan |
| Northern China | Beijing, Shaanxi, Liaoning |

population in China, 20% of the universities in the sample were selected from that region. This approach ensured the fairness and scientific accuracy of the sample distribution.

**Random selection of students**: From each selected university, participants were randomly drawn using a random number generator from student rosters spanning first-year to fourth-year undergraduates. This process ensured that every student had an equal chance of selection, minimizing selection bias and achieving balanced representation in terms of gender, academic year, and field of study.

**Exclusion of certain regions**: Due to logistical constraints and geographical complexities, some regions were excluded from the sampling frame. For instance, Xinjiang and Tibet were excluded because of their vast geographical areas, remote locations, and diverse ethnic compositions, which complicated data collection. Taiwan was not included due to additional challenges related to research permissions and participant cooperation. While the exclusion of these regions may somewhat limit the generalizability of the findings, the included regions sufficiently represent the overall characteristics of China's college student population, providing broad reliability and representativeness for the study's conclusions.

### 3.1.3 Data collection methods.
To ensure the validity and reliability of the data, a standardized data collection process was strictly followed, as outlined below:

**Training of data collectors**: Before the distribution of questionnaires, all members of the research team underwent systematic training. The training covered the clarification of research objectives, the importance of random sampling, and the standardized procedures for distributing and collecting questionnaires. This unified training ensured consistency among all researchers throughout the data collection process.

**Questionnaire distribution**: The distribution of questionnaires was scheduled on regular school days and was supervised by the research team to ensure that students could complete the questionnaires independently in an environment free from external interference. To maintain the integrity and accuracy of the data, completed questionnaires were collected immediately after they were filled out.

**Confidentiality and informed consent**: Confidentiality and Informed Consent: This study strictly adhered to the Declaration of Helsinki and relevant national and institutional ethical guidelines. Ethical approval was obtained from the Ethics Committee of Chengdu Sport University (Approval Number: CTYLL2024010). All participants were adults and provided verbal informed consent before participating in the study, confirming their understanding of the study's content and voluntary involvement. During the consent process, researchers provided each participant with a detailed explanation of the study's purpose, procedures, and their rights, ensuring that participants had ample time to ask questions. During data processing, all responses were fully anonymized, ensuring the privacy and confidentiality of participants' data were thoroughly protected.

### 3.1.4 Data processing.
The survey was conducted over a five-month period, starting on April 1, 2024, and concluding on September 1, 2024. A total of 2,200 questionnaires were distributed during this time. To ensure data accuracy and research rigor, all returned questionnaires underwent meticulous screening. Invalid questionnaires, including those with missing answers, errors, or fixed response patterns, were excluded from the dataset. Ultimately, 2,036 valid questionnaires were retained, resulting in a high effective response rate of 92.55%. During the screening process, predefined criteria were strictly applied to eliminate incomplete responses or inconsistencies in answers, ensuring the high quality and integrity of the data. Detailed information about the respondents is presented in Table 2.

## 3.2 Measurement

The measurement of physical exercise utilized a scale developed by Liang Deqing et al. [70], comprising 3 items. Responses were rated on a 5-point Likert scale ranging from 1 (very poor) to 5 (very good), evaluating the level of individuals' physical activity. This scale has demonstrated excellent reliability and validity in previous research, with an internal consistency reliability (Cronbach's α) of 0.885 in this study, further confirming its reliability.

**Table 2. The sample information.**

| Basic information | Category | Frequency | Percentage | Cumulative percentage |
|---|---|---|---|---|
| Gender | Male | 1011 | 49.66% | 49.66% |
| | Female | 1025 | 50.34% | 100% |
| Grade | Freshman (Year 1) | 506 | 24.85% | 24.85% |
| | Sophomore (Year 2) | 503 | 24.71% | 49.56% |
| | Junior (Year 3) | 501 | 24.61% | 74.17% |
| | Senior (Year 4) | 526 | 25.83% | 100% |

The assessment of feelings of inferiority employed a scale developed by Wang Lei [71], consisting of 36 items across five dimensions: self-esteem, social confidence, learning ability, appearance, and physical ability. The scale also used a 5-point Likert format, from 1 (never) to 5 (always). Widely recognized for its structural validity and reliability, this scale is an essential tool for evaluating feelings of inferiority. In this study, the scale achieved an internal consistency reliability (Cronbach's α) of 0.963, indicating high reliability in measurement.

Social support was measured using a scale developed by Jiang Qianjin [72], comprising 12 items divided into three dimensions: family support, friend support, and support from others. Responses were rated on a 5-point Likert scale from 1 (completely disagree) to 5 (completely agree), providing a comprehensive evaluation of an individual's level of social support. This scale has shown strong reliability and validity in various studies and achieved an internal consistency reliability (Cronbach's α) of 0.887 in this study, confirming the stability and credibility of its measurements.

Table 3 summarizes the measurement scales used in this study.

### 3.3 Data analysis procedures

**3.3.1 Common method bias.** To evaluate potential common method bias in the data, this study utilized Harman's single-factor test through principal component analysis. All questionnaire items were loaded into an unrotated factor solution to examine the proportion of variance explained by a single factor. If the single factor accounted for more than 40% of the total variance, it could indicate the presence of common method bias, which might threaten the validity of the study.

**3.3.2 Descriptive statistics.** This study conducted statistical analyses of the means and standard deviations for the three key variables: physical exercise, social support, and feelings of inferiority among college students. This process aimed to reveal the central tendencies and variability characteristics of the sample, providing a comprehensive depiction of the sample's overall profile and laying a solid foundation for subsequent analyses.

**3.3.3 Internal consistency reliability.** This study utilized Cronbach's alpha coefficient to evaluate the internal consistency of each dimension, ensuring the reliability of the scales in measuring their intended constructs. Generally, an alpha coefficient of 0.70 or above is considered acceptable, while a value exceeding 0.80 indicates high internal consistency. This analysis confirmed the stability and coherence of the scales, providing a robust foundation for the scientific validity and reliability of the study's results.

**Table 3. Scales used in this study.**

| Scale | Author (Year) | Item quantity | Scoring | Dimensions |
|---|---|---|---|---|
| Physical exercise | Deqing Liang (1994) | 3 | 5 | Exercise intensit, Duratio, exercise Frequency |
| Social support | Qianjin Jiang (2001) | 12 | 5 | Family, Friends, Others |
| Feelings of inferiority | Lei Wang (1999) | 36 | 5 | Self-esteem, Social confidence, Learning ability, Appearance, Physical fitness |

**3.3.4 Confirmatory factor analysis (CFA).** To validate the measurement model's fit, this study conducted confirmatory factor analysis (CFA) in two stages. First, CFA was performed individually for the three constructs—physical exercise, social support, and feelings of inferiority—to independently test whether their factor structures aligned with theoretical expectations. The evaluation of model fit was based on the following widely accepted criteria: $\chi^2/df < 3.0$, CFI (Comparative Fit Index) > 0.90, TLI (Tucker-Lewis Index) > 0.90, RMSEA (Root Mean Square Error of Approximation) < 0.08, and SRMR (Standardized Root Mean Square Residual) < 0.08. These thresholds ensured that the factor models achieved an acceptable level of fit based on recognized standards.

Second, to evaluate the discriminant validity among constructs, multiple CFA models were tested, comparing the three-factor model with alternative combined models. A superior fit of the three-factor model indicated that the constructs exhibited good discriminant validity, thereby supporting the independence of the dimensions.

**3.3.5 Correlation analysis.** This study conducted correlation analysis to examine the relationships among physical exercise, social support, and feelings of inferiority in college students, with the significance level set at $p < 0.05$ to determine statistically significant correlations between the variables. By analyzing the magnitude and direction of the correlation coefficients, preliminary validation and support were provided for the hypothesized model.

**3.3.6 Evaluation of structural model fit.** This study evaluated the overall fit of the structural model using multiple fit indices to verify whether the structural equation model (SEM) accurately reflected the data characteristics. The evaluation criteria included $\chi^2/df < 3.0$, CFI > 0.90, TLI > 0.90, SRMR < 0.08, and RMSEA < 0.08. These indices collectively ensured that the model's fit met widely accepted scientific standards, providing a solid foundation for validating the research hypotheses and the accuracy of the proposed path relationships.

**3.3.7 Path analysis.** This study conducted path analysis within the framework of structural equation modeling (SEM) to systematically examine the direct and indirect effects among the variables. The significance of path coefficients ($\beta$ values) was determined at $p < 0.05$. The analysis focused on the direct impact of physical exercise on feelings of inferiority while also exploring the indirect pathway mediated by social support. Through rigorous examination of these path relationships, the study further elucidated the interactions between variables, providing empirical support for the research hypotheses.

**3.3.8 Effect size testing.** This study evaluated the effect sizes of direct and indirect effects using standardized coefficients and applied a bootstrapping method with 2,000 resamples to compute bias-corrected confidence intervals for indirect effects. If the 95% confidence interval did not include zero, the mediation effect was deemed statistically significant. This method provided high reliability for the precise testing of mediation effects and established a solid foundation for the scientific validity of the study's conclusions.

**3.3.9 Invariance testing of structural equation models across genders.** To examine the consistency of the model across genders, this study conducted a structural invariance analysis. Nested models were compared by sequentially introducing constraints on measurement weights, structural weights, structural covariances, and structural residuals to evaluate the model's applicability for both male and female students. If the ΔCFI value was less than 0.01, the model was deemed to have structural invariance, indicating its applicability and stability across different gender groups.

## 4 Result

### 4.1 Common method bias and multicollinearity test

To assess the potential influence of common method bias in the study, Harman's single-factor test was performed. All variables were subjected to an unrotated principal component analysis (PCA) to evaluate the variance explained by a single factor. The results revealed the extraction of 9 factors with eigenvalues greater than 1, collectively accounting for a substantial proportion of the variance. Notably, the first factor explained 35.738% of the total variance, which is below the critical threshold of 40%. This finding indicates that common method bias is unlikely to significantly influence the results of this study.

## 4.2 Descriptive statistics, reliability, and construct validity of the measurement model

The descriptive statistics, internal consistency reliability, and confirmatory factor analysis (CFA) results for the key constructs—physical exercise, social support, and feelings of inferiority—are summarized in Table 4. The results show that:

**Descriptive statistics:** The mean (M) and standard deviation (SD) values indicate the general tendencies and variability for each construct. For instance, feelings of inferiority had a mean of 3.30 (SD = 0.66), suggesting a moderate level among participants.

**Reliability:** Cronbach's alpha (α) values for all constructs exceeded the recommended threshold of 0.70, indicating high internal consistency. Specifically, the reliability for physical exercise was 0.885, for social support was 0.887, and for feelings of inferiority was 0.963.

**Construct validity:** The CFA results provided evidence for the construct validity of the measurement model. Both the social support and feelings of inferiority constructs achieved excellent model fit indices, with Comparative Fit Index (CFI) values exceeding 0.980, Tucker-Lewis Index (TLI) values above 0.976, and Root Mean Square Error of Approximation (RMSEA) values within the acceptable range (0.047 and 0.027, respectively). These results confirm that the measurement model is valid and reliable.

As shown in Table 5, the fit indices for the three-factor model ($\chi^2$/df = 5.36, CFI = 0.985, TLI = 0.979, SRMR = 0.030, RMSEA = 0.046) were superior to those for the two-factor and one-factor models. The significant improvement in fit indices ($\Delta\chi^2$) supports the distinctiveness of the three constructs—physical exercise (PEX), social support (SSU), and feelings of inferiority (FOI). This finding establishes strong discriminant validity for the measurement model, as the three-factor model better represents the data compared to the alternative combined models.

In conclusion, the descriptive statistics, reliability analysis, and CFA results validate the robustness of the constructs and the appropriateness of the measurement model for the subsequent analyses.

## 4.3 Correlation analysis among key variables

The relationships among physical exercise (PEX), social support (SSU), and feelings of inferiority (FOI) were examined using Pearson correlation analysis. The results are visualized in Fig 2, which presents the correlation matrix of the key variables.

**Table 4. Descriptive Statistics, Reliability, and CFA Fit Indices for Key Variables.**

| Variable | M | SD | α | CFI | TLI | SRMR | RMSEA (90%CI) |
|---|---|---|---|---|---|---|---|
| Physical exercise | 28.97 | 31.97 | 0.885 | – | – | – | – |
| Social support | 3.41 | 0.65 | 0.887 | 0.981 | 0.976 | 0.027 | 0.047(0.042-0.052) |
| Feelings of inferiority | 3.30 | 0.66 | 0.963 | 0.982 | 0.980 | 0.019 | 0.027(0.025-0.028) |

**Table 5. Comparative fit indices for alternative factor structures of key constructs.**

| Model | Factor | $\chi^2$ | df | $\Delta\chi^2$ $\Delta$(df) | CFI | TLI | SRMR | RMSEA (90%CI) |
|---|---|---|---|---|---|---|---|---|
| Three-factor model | PEX, SSU, FOI | 134.116 | 25 | – | 0.985 | 0.979 | 0.030 | 0.046 (0.039-0.054) |
| Two-factor model | PEX+SSU, FOI | 220.140 | 26 | 86.024 (1) | 0.973 | 0.963 | 0.042 | 0.061 (0.053-0.068) |
| One-factor model | PEX+SSU+FOI | 741.643 | 27 | 607.527 (2) | 0.902 | 0.870 | 0.070 | 0.114 (0.107-0.121) |

PEX, Physical exercise. SSU, Social support. FOI, Feelings of inferiority. All $\Delta\chi^2$ passed the significance test at 0.05 level.

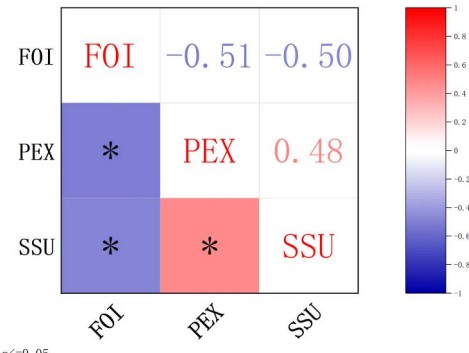

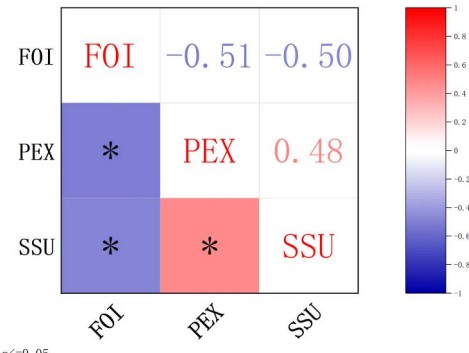

* p<=0.05

**Fig 2. Correlation Matrix of Key Variables: Physical Exercise (PEX), Social Support (SSU), and Feelings of Inferiority (FOI).** *p<0.05.*

Key findings include

**Physical exercise (PEX) and feelings of inferiority (FOI):** A significant negative correlation was observed between physical exercise and feelings of inferiority (r=-0.51, p<0.05), indicating that higher levels of physical exercise are associated with lower levels of feelings of inferiority.

**Physical exercise (PEX) and social support (SSU):** Physical exercise showed a significant positive correlation with social support (r=0.48, p<0.05), suggesting that individuals who engage in more physical exercise tend to experience greater social support.

**Social support (SSU) and feelings of inferiority (FOI):** A significant negative correlation was found between social support and feelings of inferiority (r=-0.50, p<0.05), indicating that higher levels of social support are linked to reduced feelings of inferiority.

These results provide preliminary support for the hypothesized relationships among the key variables and validate the conceptual framework of this study. The significant correlations suggest that physical exercise and social support play crucial roles in mitigating feelings of inferiority among college students. Additionally, the findings highlight the mediating potential of social support in the relationship between physical exercise and feelings of inferiority, which will be further explored in subsequent analyses.

### 4.4 Test results of mediation effects

The results of the structural equation model are summarized as follows. According to Table 6, the model achieved excellent fit indices, indicating that the hypothesized framework fits the data well. The fit indices include $\chi^2/df=5.365$, CFI=0.985, TLI=0.979, SRMR=0.030, and RMSEA=0.046 (90% CI: 0.039–0.054), all within the acceptable range.

The relationships among physical exercise, social support, and feelings of inferiority were examined through path analysis. As shown in Fig 3, physical exercise had a significant direct negative effect on feelings of inferiority ($\beta$=-0.259, p<0.001), which supports Hypothesis 1 (H1). This finding confirms that as levels of physical exercise increase, feelings of inferiority among college students decrease.

**Table 6. Questionnaire model fitting indicators.**

| Model Fit | $\chi^2/df$ | CFI | TLI | SRMR | RMSEA (90%CI) |
|---|---|---|---|---|---|
| Model | 5.365 | 0.985 | 0.979 | 0.030 | 0.046 (0.039, 0.054) |

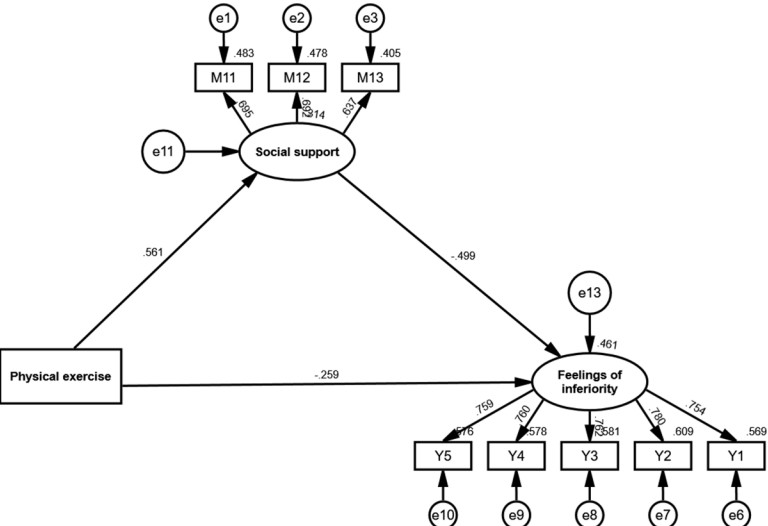

**Fig 3. Structural equation model depicting the relationships between physical exercise, social support, and feelings of inferiority.**

Additionally, physical exercise showed a significant positive effect on social support (β = 0.561, p < 0.001), validating Hypothesis 2 (H2). This suggests that engaging in physical exercise can enhance the social support that individuals perceive, providing a beneficial platform for emotional and interpersonal connections. Social support, in turn, demonstrated a significant negative effect on feelings of inferiority (β = -0.499, p < 0.001), supporting Hypothesis 3 (H3). This indicates that higher levels of social support are associated with a reduction in feelings of inferiority.

Furthermore, mediation analysis results in Table 7 confirmed that social support mediates the relationship between physical exercise and feelings of inferiority. The indirect effect of physical exercise on feelings of inferiority through social support was significant (β = -0.280, p < 0.001), as the 95% bootstrap confidence interval (-0.321, -0.243) did not include zero. The indirect effect accounted for 51.95% of the total effect, while the direct effect accounted for 48.05%. The total effect of physical exercise on feelings of inferiority was β = -0.539, p < 0.001, emphasizing the dual pathways—direct and mediated—through which physical exercise influences feelings of inferiority. These results provide strong evidence supporting Hypothesis 4 (H4), which posits that social support mediates the relationship between physical exercise and feelings of inferiority.

In conclusion, Table 7 underscores the importance of mediation in this framework. It highlights that physical exercise significantly reduces feelings of inferiority among college students, not only directly but also indirectly through the enhancement of social support. This dual mechanism demonstrates the critical role of social support as a psychological resource in alleviating negative emotional states and maximizing the mental health benefits of physical exercise.

**Table 7. Total, direct and indirect effects in the multiple mediator model.**

| Path | Estimated effect | Boot SE | P | Boot LLCI | Boot ULCI | Ratio |
|---|---|---|---|---|---|---|
| Direct effect | | | | | | |
| PEX→FOI | -0.259 | 0.027 | 0.001 | -0.309 | -0.204 | 48.05% |
| Indirect effects | | | | | | |
| PEX→SSU→FOI | -0.280 | 0.020 | 0.001 | -0.321 | -0.243 | 51.95% |
| Total effect | -0.539 | 0.017 | 0.000 | -0.569 | -0.506 | – |

PEX, Physical exercise. SSU, Social support. FOI, Feelings of inferiority. Boot LLCI, the lower bound of the 95% confidence interval. Boot ULCI, the upper limit of the 95% confidence interval (Percentile Bootstrap Method with Bias Correction). The Bootstrap sample size is set at 2000.

## 4.5  Testing for structural invariance across gender

To examine whether the structural equation model is applicable across male and female college students, a multi-group analysis was conducted to test for structural invariance. The results are presented in Table 8 and include fit indices for different levels of constraint: unconstrained, measurement weights, structural weights, structural covariances, and structural residuals.

The unconstrained model fit the data well, with $\chi^2/df = 3.180$, CFI = 0.985, TLI = 0.979, SRMR = 0.031, and RMSEA = 0.033 (90% CI: 0.027–0.038). Subsequent models added increasing constraints to test measurement and structural invariance. The changes in CFI (ΔCFI) and TLI (ΔTLI) were minimal across all levels of constraint, with ΔCFI ≤ 0.002 and ΔTLI ≤ 0.002, meeting the widely accepted thresholds (ΔCFI < 0.01, ΔTLI < 0.01) for invariance. These results indicate that the factor loadings, structural paths, and covariances do not differ significantly between male and female students.

Specifically

**Measurement weights invariance:** When equality constraints were imposed on the factor loadings, the model fit remained excellent (CFI = 0.984, ΔCFI = -0.001; TLI = 0.981, ΔTLI = +0.002), indicating that both genders interpret the latent constructs in a similar manner.

**Structural weights invariance:** Adding constraints on the structural paths showed no significant changes in model fit indices (CFI = 0.984, ΔCFI = -0.001; TLI = 0.981, ΔTLI = +0.002), supporting the equality of structural relationships across genders.

**Structural covariances and residuals invariance:** When constraints were applied to covariances and residuals, the model continued to fit well, with minimal changes in CFI and TLI values (ΔCFI ≤ 0.002, ΔTLI ≤ +0.001), confirming invariance at these levels.

In summary, the invariance testing results demonstrate that the structural equation model is stable and consistent across male and female college students. The findings suggest that the relationships among physical exercise, social support, and feelings of inferiority are not moderated by gender, highlighting the generalizability of the proposed model across different gender groups. This invariance further strengthens the validity and reliability of the study's conclusions.

# 5  Discussion

This study aimed to explore the relationships among physical exercise, social support, and feelings of inferiority in college students, with a particular focus on the mediating role of social support. The findings not only confirm the hypothesized relationships but also shed light on the intricate mechanisms through which physical exercise alleviates feelings of inferiority. This section delves into the implications of these findings, contextualizes them within the broader literature, and highlights their theoretical and practical significance.

**Table 8.  Testing for structural invariance across gender.**

| | $\chi^2/df$ | CFI | △CFI | TLI | △TLI | SRMR | RMSEA (90%CI) |
|---|---|---|---|---|---|---|---|
| Unconstrained | 3.180 | 0.985 | – | 0.979 | – | 0.031 | 0.033 (0.027-0.038) |
| Measurement weights | 2.983 | 0.984 | -0.001 | 0.981 | +0.002 | 0.036 | 0.031 (0.026-0.037) |
| Structural weights | 2.935 | 0.984 | -0.001 | 0.981 | +0.002 | 0.036 | 0.031 (0.026-0.036) |
| Structural covariances | 3.108 | 0.983 | -0.002 | 0.979 | 0 | 0.033 | 0.032 (0.027-0.037) |
| Structural residuals | 3.023 | 0.983 | -0.002 | 0.980 | +0.001 | 0.033 | 0.032 (0.026-0.037) |

### 5.1 Physical exercise as a direct buffer against feelings of inferiority

The results provide compelling evidence that physical exercise has a significant direct negative effect on feelings of inferiority. This aligns with prior research suggesting that physical exercise enhances self-perception and physical self-esteem, thereby counteracting the negative self-evaluations that characterize inferiority [16,34]. Importantly, this study extends existing findings by demonstrating that the effect is robust among college students, a population particularly vulnerable to feelings of inferiority due to academic pressure and the demands of social comparison.

A key reason for this relationship lies in the psychological benefits of physical activity. Beyond physical health, exercise fosters a sense of achievement and mastery. Engaging in structured physical activities allows students to develop self-discipline, set and achieve goals, and experience tangible improvements in physical performance, all of which contribute to improved self-worth [19]. Furthermore, competitive sports or fitness activities often provide an opportunity to receive external validation for physical abilities, further reinforcing positive self-perceptions.

This direct link between physical exercise and feelings of inferiority has important implications for intervention strategies. Physical activity should not only be promoted for its physical health benefits but also as an accessible and effective tool for improving psychological resilience in students. Universities and mental health professionals should consider leveraging physical activity programs as part of broader strategies to address negative emotional states among students.

### 5.2 Social support as a mediating mechanism

Social support emerged as a critical mediating factor in the relationship between physical exercise and feelings of inferiority. The findings demonstrate that physical exercise significantly enhances social support, which in turn reduces feelings of inferiority. This mediation pathway accounted for 51.95% of the total effect, underscoring the substantial role of social support in amplifying the psychological benefits of physical exercise.

This study supports the idea that physical exercise, particularly in social or team settings, fosters interaction and cooperation among participants. Team sports, group fitness classes, and even casual recreational activities provide a platform for building trust, collaboration, and mutual encouragement [20,29]. These interactions lead to increased emotional and practical support, fulfilling individuals' psychological needs for belonging and recognition. Additionally, receiving recognition and encouragement during physical activities can significantly enhance social bonds, reducing feelings of loneliness and isolation that often accompany inferiority.

Moreover, social support acts as a buffer against the cognitive distortions that drive feelings of inferiority. The encouragement and positive feedback received from peers and coaches help individuals challenge their negative self-evaluations and adopt more balanced, positive perspectives [8,41]. For instance, participants who initially feel inadequate in physical performance may gradually build confidence through repeated encouragement and successful experiences, weakening the negative self-concept at the root of inferiority.

This finding is particularly significant as it demonstrates the dual role of physical exercise in addressing inferiority—not only as an individual activity that builds personal competence but also as a social platform that fosters critical interpersonal connections.

### 5.3 Gender invariance: Consistency across male and female students

The invariance testing revealed that the relationships among physical exercise, social support, and feelings of inferiority are consistent across genders. This finding has two important implications. First, it highlights the universal applicability of physical exercise as a psychological intervention. Both male and female students benefit from the direct and indirect effects of physical exercise on reducing feelings of inferiority. This consistency suggests that gender-specific approaches to promoting physical activity may not be necessary, as the psychological mechanisms underpinning its benefits are similar for both genders.

Second, the results suggest that universities can design inclusive physical activity programs that cater equally to male and female students. Programs emphasizing both individual activities and team sports can be effective in fostering social support and reducing inferiority, regardless of gender [19,30]. However, future research could further explore whether specific types of physical activities (e.g., individual versus team-based, high- versus moderate-intensity) yield differential benefits across genders.

### 5.4 Theoretical contributions

This study makes several important contributions to the literature. First, it advances understanding of the role of physical exercise in alleviating negative emotional states such as feelings of inferiority. While prior research has extensively examined the benefits of physical exercise on mental health, this study specifically identifies inferiority as a unique construct affected by exercise, filling a gap in the existing literature.

Second, the findings provide empirical support for the mediating role of social support, offering a nuanced understanding of how physical exercise exerts its psychological benefits. This underscores the importance of considering both individual and social pathways when examining the mental health outcomes of physical activity. The dual pathway model proposed in this study—direct effects through improved self-perception and indirect effects through social support—provides a comprehensive framework for future research.

Third, the study's robust methodological approach, including the use of structural equation modeling and multi-group invariance testing, enhances the validity and generalizability of the findings. The demonstration of gender invariance further strengthens the applicability of the proposed model across diverse student populations.

### 5.5 Practical implications

The practical implications of this study are substantial. Universities should prioritize physical activity programs as part of their mental health initiatives. Specifically, team-based sports and group fitness activities should be emphasized, as they provide both physical and social benefits [14]. Such programs can be integrated into campus wellness initiatives or offered as part of orientation activities to help students build social connections early in their college experience.

Additionally, the findings highlight the importance of fostering supportive social environments within physical activity settings. Coaches, fitness instructors, and peer leaders play a crucial role in providing encouragement and positive feedback, which can amplify the mental health benefits of physical activity [20]. Universities should train these individuals to recognize and address the psychological needs of participants, particularly those struggling with feelings of inferiority.

Finally, the universal applicability of the findings suggests that these interventions can be widely implemented across diverse student populations. However, culturally tailored approaches may be necessary to address unique challenges faced by students in different regions or educational contexts.

## 6 Limitations and future research

Despite its strengths, this study has several limitations. First, its cross-sectional design limits the ability to draw causal inferences. Longitudinal studies are needed to confirm the temporal relationships among the variables and to examine whether the effects of physical exercise and social support on feelings of inferiority are sustained over time. Second, the study was conducted in a Chinese college student population, which may limit the generalizability of the findings to other cultural contexts. Future research should replicate this study in diverse cultural and educational settings to enhance external validity.

Additionally, this study focused on social support as the primary mediator. Future research could explore other potential mediators, such as emotional regulation, self-efficacy, or social identity, to provide a more comprehensive understanding of the mechanisms underlying the effects of physical exercise. Furthermore, examining the effects of different types of physical activities (e.g., aerobic versus resistance training, team-based versus individual activities) could yield more specific recommendations for interventions.

## 7 Conclusion

This study underscores the significant impact of physical exercise on reducing feelings of inferiority among college students, both directly and indirectly through social support. By fostering a sense of achievement and enhancing interpersonal connections, physical exercise serves as a powerful tool for improving students' mental health. The findings highlight the critical role of social support as a mediator, emphasizing the need to integrate social dimensions into physical activity interventions. Additionally, the consistency of the model across genders suggests that these benefits are universally applicable. These results provide a strong foundation for evidence-based interventions that leverage physical activity to enhance the psychological well-being of college students, ultimately promoting healthier and more resilient student populations.

## Supporting information

**S1 Data. English version of the supporting data.**
(ZIP)

## Acknowledgments

The authors thank all individuals who participated in this study.

## Author contributions

**Data curation:** Weisong Chen.

**Formal analysis:** Huan Lei, Minghuan Tang, hongshen wang.

**Investigation:** Huan Lei, Bo Peng, Ting Yu.

**Methodology:** Huan Lei, Bo Peng, Minghuan Tang, Weisong Chen, Ting Yu.

**Project administration:** hongshen wang.

**Writing – original draft:** Huan Lei, Bo Peng, Minghuan Tang, Weisong Chen, hongshen wang, Ting Yu.

**Writing – review & editing:** Huan Lei, Bo Peng, Minghuan Tang, Weisong Chen, hongshen wang, Ting Yu.

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
