## [Decision Letter · Decision Letter 0]

23 Jan 2025

PONE-D-24-56284A study on the relationship between college students' physical exercise and feelings of inferiority: the mediating effect of social supportPLOS ONE

Dear Dr. wang,

Thank you for submitting your manuscript to PLOS ONE. After careful consideration, we feel that it has merit but does not fully meet PLOS ONE’s publication criteria as it currently stands. Therefore, we invite you to submit a revised version of the manuscript that addresses the points raised during the review process.

We look forward to receiving your revised manuscript.

Kind regards,

Maria José Nogueira, Ph.D.

Academic Editor

PLOS ONE

Journal Requirements:

2. In the ethics statement in the Methods, you have specified that verbal consent was obtained. Please provide additional details regarding how this consent was documented and witnessed, and state whether this was approved by the IRB.

4. We note that your Data Availability Statement is currently as follows: 

“The original contributions presented in the study are included in the article/supplementary material, further inquiries can be directed to the corresponding author/s.”

**Additional Editor Comments:**

Congratulations on the work

This work is an important contribution to improving students’ mental health.

However, with regard to the presentation of results (tables, table titles, diagrams), the information can be presented in a more attractive way, which facilitates its analysis.

Reviewers' comments:

Reviewer's Responses to Questions

**Comments to the Author**

1. Is the manuscript technically sound, and do the data support the conclusions?

Reviewer #1: Yes

2. Has the statistical analysis been performed appropriately and rigorously? 

Reviewer #1: Yes

3. Have the authors made all data underlying the findings in their manuscript fully available?

Reviewer #1: Yes

4. Is the manuscript presented in an intelligible fashion and written in standard English?

Reviewer #1: Yes

5. Review Comments to the Author

Reviewer #1: This paper proposes an investigation into the physical exercise habits of university students and their feelings of inferiority, focusing on the mediating role of social support. The topic under study is very pertinent and links physical activity and social interactions and how these contribute to the well-being of university students. It provides a good theoretical framework and has a methodology appropriate to the topic under study. The conclusions are relevant and point to ways for the development and deepening of the topic.

6. PLOS authors have the option to publish the peer review history of their article (what does this mean? ). If published, this will include your full peer review and any attached files.

**Do you want your identity to be public for this peer review?** For information about this choice, including consent withdrawal, please see our Privacy Policy .

Reviewer #1: No

---

## [Author Response · Author response to Decision Letter 0]

5 Mar 2025

Response to Reviewers

We sincerely thank the editor and reviewers for their detailed and constructive comments on our manuscript. Your insightful feedback has been invaluable in helping us improve the quality, rigor, and clarity of our manuscript. We have carefully reviewed all the comments and have made substantial revisions to address the concerns raised. Below, we provide a point-by-point response to each of the reviewers' comments, detailing the revisions we have made.

Once again, we deeply appreciate your time, expertise, and constructive suggestions, all of which have significantly strengthened our study. We hope that the revised manuscript meets your expectations.

Editor

Comment 1:File naming.

Reviewer’s Comment:Please ensure that your manuscript meets PLOS ONE's style requirements, including those for file naming. The PLOS ONE style templates can be found at https://journals.plos.org/plosone/s/file?id=wjVg/PLOSOne_formatting_sample_main_body.pdf and https://journals.plos.org/plosone/s/file?id=ba62/PLOSOne_formatting_sample_title_authors_affiliations.pdf

Response:We would like to sincerely thank the reviewer for pointing out the importance of adhering to PLOS ONE's style requirements, including those for file naming. We have carefully reviewed the provided style templates and have made the necessary adjustments to ensure that our manuscript fully complies with PLOS ONE's formatting guidelines. We appreciate your valuable feedback, which has helped us improve the quality and presentation of our submission.

Revisions Made:

We would like to express our sincere gratitude to the reviewer for pointing out the importance of adhering to PLOS ONE's style requirements, including those for file naming. In response, we have thoroughly reviewed the provided style templates and made the necessary adjustments to ensure that our manuscript fully complies with PLOS ONE's formatting guidelines. We have updated the manuscript to reflect these changes, including the correct file naming conventions. Thank you for your valuable feedback, which has helped us improve the presentation of our work.

Specific Modifications:

1:The title and main body fonts were checked to ensure they comply with PLOS ONE's style requirements.

2:The title and main body fonts were then modified to match the font and size specified by PLOS ONE.

Comment 2:Details of consent obtained

Reviewer’s Comment:In the ethics statement in the Methods, you have specified that verbal consent was obtained. Please provide additional details regarding how this consent was documented and witnessed, and state whether this was approved by the IRB.

Response:We would like to sincerely thank the reviewer for their valuable feedback regarding the ethics statement. In response to the comment, we have revised the section to provide additional details on how verbal consent was obtained, documented, and witnessed. We also clarified that the study was approved by the Ethics Committee of Chengdu Sport University (Approval Number: CTYLL2024010). These revisions ensure a more comprehensive description of the consent process, as requested. We appreciate your careful review and believe these modifications strengthen the manuscript.

Revisions Made:

Confidentiality and informed consent:Confidentiality and Informed Consent: This study strictly adhered to the Declaration of Helsinki and relevant national and institutional ethical guidelines. Ethical approval was obtained from the Ethics Committee of Chengdu Sport University (Approval Number: CTYLL2024010). All participants were adults and provided verbal informed consent before participating in the study, confirming their understanding of the study's content and voluntary involvement. During the consent process, researchers provided each participant with a detailed explanation of the study's purpose, procedures, and their rights, ensuring that participants had ample time to ask questions. During data processing, all responses were fully anonymized, ensuring the privacy and confidentiality of participants' data were thoroughly protected.

Specific Modifications:

1:Detailed the process of obtaining verbal consent. We further clarified the process of obtaining verbal consent in the ethics statement, explaining that verbal consent was obtained after the researcher provided a detailed explanation of the study’s purpose, procedures, and the participants' rights.

2:Added details on how verbal consent was documented and witnessed. We described how verbal consent was documented and emphasized that it was obtained voluntarily by the participants after understanding the study’s content. We also clarified that the verbal consent was witnessed by the researchers.

3:Clarified IRB approval. We clearly stated that the study was approved by the Ethics Committee of Chengdu Sport University (Approval Number: CTYLL2024010) to ensure compliance with academic and ethical standards.

Comment 3:Data sharing

Reviewer’s Comment:We note that you have indicated that there are restrictions to data sharing for this study. For studies involving human research participant data or other sensitive data, we encourage authors to share de-identified or anonymized data. However, when data cannot be publicly shared for ethical reasons, we allow authors to make their data sets available upon request. For information on unacceptable data access restrictions, please see http://journals.plos.org/plosone/s/data-availability#loc-unacceptable-data-access-restrictions.

a)If there are ethical or legal restrictions on sharing a de-identified data set, please explain them in detail (e.g., data contain potentially identifying or sensitive patient information, data are owned by a third-party organization, etc.) and who has imposed them (e.g., a Research Ethics Committee or Institutional Review Board, etc.). Please also provide contact information for a data access committee, ethics committee, or other institutional body to which data requests may be sent.

b)b) If there are no restrictions, please upload the minimal anonymized data set necessary to replicate your study findings to a stable, public repository and provide us with the relevant URLs, DOIs, or accession numbers. Please see http://www.bmj.com/content/340/bmj.c181.long for guidelines on how to de-identify and prepare clinical data for publication. For a list of recommended repositories, please see https://journals.plos.org/plosone/s/recommended-repositories. You also have the option of uploading the data as Supporting Information files, but we would recommend depositing data directly to a data repository if possible.

Response:We sincerely thank the reviewer for their valuable feedback regarding data sharing. In response to the reviewer's comment, we have made the necessary revisions in the manuscript. Additionally, we have uploaded the data as Supporting Information to comply with the journal's requirements. We appreciate the reviewer's suggestion, which has helped improve the transparency and clarity of the manuscript regarding data usage.

Revisions Made:

12 Supporting information

S1 Empirical data and original scale. (zip)

Specific Modifications:

We have made modifications to the data usage section of the manuscript. We have updated the data section to "12 Supporting information: S1 Empirical data and original scale. (zip)" to ensure the transparency of data sharing and compliance with the journal's requirements.

Comment 4:Raw data

Reviewer’s Comment:We note that your Data Availability Statement is currently as follows:

“The original contributions presented in the study are included in the article/supplementary material, further inquiries can be directed to the corresponding author/s.”

The values behind the means, standard deviations and other measures reported;

The values used to build graphs;

The points extracted from images for analysis.

Response:We sincerely thank the reviewer for their valuable feedback on the Data Availability Statement. In response to the reviewer's comment, we have reviewed and updated the data usage to ensure compliance with the requirements for sharing the minimal data set. We confirm that the submitted materials include all the raw data required to replicate the study's results, including the values behind the reported means, standard deviations, and other measures, as well as the data used to build the graphs and analyze images. To comply with PLOS ONE’s guidelines, we have uploaded the data as Supporting Information and made it available for review. We appreciate the reviewer’s suggestions, which have helped us improve the transparency and compliance of our submission.

Revisions Made:

12 Supporting information

S1 Empirical data and original scale. (zip)

Specific Modifications:

We have made modifications to the data usage section of the manuscript. We have updated the data section to "12 Supporting information: S1 Empirical data and original scale. (zip)" to ensure the transparency of data sharing and compliance with the journal's requirements.

Comment 5: Corresponding author ORCID iD.

Reviewer’s Comment: PLOS requires an ORCID iD for the corresponding author in Editorial Manager on papers submitted after December 6th, 2016. Please ensure that you have an ORCID iD and that it is validated in Editorial Manager. To do this, go to ‘Update my Information’ (in the upper left-hand corner of the main menu), and click on the Fetch/Validate link next to the ORCID field. This will take you to the ORCID site and allow you to create a new iD or authenticate a pre-existing iD in Editorial Manager.

Response: We would like to sincerely thank the reviewer for pointing out the requirement for an ORCID iD for the corresponding author. In response to this comment, we have ensured that the corresponding author's ORCID iD has been created and validated in Editorial Manager. We followed the provided instructions and updated the ORCID iD in the system. We appreciate the reviewer’s helpful suggestion, which has allowed us to meet the journal's requirements for ORCID iD validation.

Revisions Made:

The ORCID iD of the corresponding author is 0009-0007-8947-0234.

Specific Modifications:

1:Created and validated the corresponding author's ORCID iD. In response to the reviewer's comment, we ensured that the corresponding author's ORCID iD was created and validated in the Editorial Manager system.

2:Updated the ORCID iD following the provided instructions. We followed the steps provided by the reviewer, went to the "Update my Information" page, clicked the "Fetch/Validate" link next to the ORCID field, and accessed the ORCID website, where we successfully created a new ORCID iD and validated it.

3:Ensured the ORCID iD complies with journal requirements. We have ensured that the ORCID iD complies with PLOS ONE's requirements and completed all necessary updates in the submission system.

Comment 6: Check references.

Reviewer’s Comment: Please review your reference list to ensure that it is complete and correct. If you have cited papers that have been retracted, please include the rationale for doing so in the manuscript text, or remove these references and replace them with relevant current references. Any changes to the reference list should be mentioned in the rebuttal letter that accompanies your revised manuscript. If you need to cite a retracted article, indicate the article’s retracted status in the References list and also include a citation and full reference for the retraction notice.

Response: We would like to sincerely thank the reviewer for their valuable feedback regarding the reference list. In response to this comment, we have carefully reviewed the entire reference list to ensure that it is complete and correct. We have confirmed that there are no retracted papers cited in the manuscript. If there were any, we would have included the rationale for citing them or replaced them with relevant current references as suggested. No changes to the reference list were necessary. We appreciate the reviewer's suggestion, which has helped ensure the accuracy and integrity of our reference list.

Comment 7: Improvement of tables, table titles, and figures.

Reviewer’s Comment: This work is an important contribution to improving students’ mental health.

However, with regard to the presentation of results (tables, table titles, diagrams), the information can be presented in a more attractive way, which facilitates its analysis.

Response: We would like to sincerely thank the reviewer for their positive feedback on the contribution of our work to improving students' mental health. In response to the comment regarding the presentation of results, we have made improvements to the tables, table titles, and diagrams. We have reformatted and enhanced the presentation to make the information clearer and more visually appealing, which we believe will facilitate better analysis and comprehension of the results. We appreciate the reviewer’s suggestion, which has greatly helped in improving the overall presentation of our manuscript.

Revisions Made:

Table 4. Descriptive Statistics, Reliability, and CFA Fit Indices for Key Variables.

Figure1. heoretical Model of the Relationship Between Physical Exercise and Feelings of Inferiority in College Students: The Mediating Role of Social Support.

Figure 2. Correlation Matrix of Key Variables: Physical Exercise (PEX), Social Support (SSU), and Feelings of Inferiority (FOI). p < 0.05.

Figure 3. Structural equation model depicting the relationships between physical exercise, social support, and feelings of inferiority.

Specific Modifications:

1:Improved the design of tables and figures. We formatted and enhanced all tables and figures to make them clearer and more visually appealing, facilitating easier understanding and analysis of the results.

2:Refined the table titles. In response to the reviewer's suggestion, we renamed several table titles to make them more accurate, concise, and better reflective of the content.

3:Adjusted the layout of figures. We reorganized the layout of the figures to make the data more intuitive, easier to compare, and more effective in displaying the results.

We sincerely appreciate your insightful comments, which have greatly enhanced the clarity, accuracy, and overall presentation of the manuscript. Thank you for your valuable feedback.

Reviewer 1

Comment 1:Supportive comments.

Reviewer’s Comment:This paper proposes an investigation into the physical exercise habits of university students and their feelings of inferiority, focusing on the mediating role of social support. The topic under study is very pertinent and links physical activity and s

---

## [Editor Report · Decision Letter 1]

10 Mar 2025

A study on the relationship between college students' physical exercise and feelings of inferiority: the mediating effect of social support

PONE-D-24-56284R1

Dear Dr. hongshen wang

We’re pleased to inform you that your manuscript has been judged scientifically suitable for publication and will be formally accepted for publication once it meets all outstanding technical requirements.

Kind regards,

Maria José Nogueira, Ph.D.

Academic Editor

PLOS ONE

Additional Editor Comments (optional):

Dear authors

The authors introduced the suggested improvements to the manuscript, which can now be accepted for publication.
---

## [Editor Report · Acceptance letter]

PONE-D-24-56284R1

PLOS ONE

Dear Dr. wang,

I'm pleased to inform you that your manuscript has been deemed suitable for publication in PLOS ONE. Congratulations! Your manuscript is now being handed over to our production team.

Kind regards,

on behalf of

Professor Maria José Nogueira

Academic Editor

PLOS ONE